



# Human influence on growing-period frosts like the early April 2021 in Central France

Robert Vautard[1], Geert Jan van Oldenborgh[2,†], Rémy Bonnet[1], Sihan Li[3], Yoann Robin[4], Sarah Kew[2], Sjoukje Philip[2], Jean-Michel Soubeyroux[4], Brigitte Dubuisson[4], Nicolas Viovy[5], Markus Reichstein [6], Friederike Otto[7], Iñaki Garcia de Cortazar-Atauri[8]

[1]Institut Pierre-Simon Laplace, CNRS, Sorbonne Université, Université de Versailles - Saint Quentin en Yvelines, France
[2]Koninklijk Nederlands Meteorologisch Instituut, de Bilt, Netherlands
[3]School of Geography and the Environment, University of Oxford
[4]Météo-France, Toulouse, France
[5]Laboratoire des Science du Climat et de l'Environnement, CEA, CNRS, Université de Versailles - Saint Quentin en Yvelines, IPSL, France
[6] Department of Biogeochemical Integration, Max-Planck Institute for Biogeochemistry, 07701 Jena, Germany
[7]Grantham Institute, Imperial College, London, United Kingdom
[8]INRAE, US AgroClim, 84914, Avignon, France
†Deceased, 12 October 2021

*Correspondence to*: Robert Vautard (Robert.vautard@lsce.ipsl.fr)

**Abstract.** In early April 2021 several days of harsh frost affected central Europe. This led to very severe damages in grapevine and fruit trees in France, in regions where young leaves had already unfolded due to unusually warm temperatures in the preceding month (march 2021. We analysed with observations and 172 climate model simulations how human-induced climate change affected this event over central France, where many vineyards are located. We found that, without human-caused climate change, such temperatures in April or later in spring would have been even lower by 1.2°C [0.75°C;1.7°C]. However, climate change also caused an earlier occurrence of bud burst, that we characterized in this study by a growing-degree-day index value. This shift leaves young leaves exposed to more winter-like conditions with lower minimum temperatures and longer nights, an effect that over-compensates the warming effect. Extreme cold temperatures occurring after the start of the growing season such as those of April 2021 are now 2°C colder [0.5°C to 3.3°C] than in pre-industrial conditions, according to observations. This observed intensification of growing-period frosts is attributable, at least in part, to human-caused climate change with each of 5 climate model ensembles used here simulating a cooling of growing-period annual temperature minima of 0.41°C [0.22°C to 0.60°C] since pre-industrial conditions. The 2021 growing-period frost event has become 50% more likely [10%-110%]. Models accurately simulate the observed warming in extreme lowest spring temperatures, but underestimate the observed trends in growing-period frost intensities, a fact that remains yet to be explained. Model ensembles all simulate a further intensification of yearly minimum temperatures occurring in the growing period for future decades, and a significant probability increase for such events of about 30% [20%-40%] in a climate with global warming of 2°C.

## 1 Introduction

Frost days and cold spells are decreasing in frequency and intensity worldwide (IPCC, 2021; van Oldenborgh et al., 2019). Yet, severe cold spells continue to pound many mid-latitude areas, due to the occasional invasion of polar air being transported well into lower latitudes as a consequence of the chaotic motion of Rossby waves. When occurring in spring, such cold events can create a range of impacts on agriculture such as in April 2021, when young leaves and flowers have started to develop in fruit trees or grapevines. The frost event which took place from 6 to 8 April 2021 was exceptional with daily minimum

temperatures below -5°C recorded in several places. In several places, such low temperatures left no chance to save grapevines and fruit trees by frost management strategies (e.g. local heating from braseros or spreading water to keep frost moderate at the surface of plants). The cold temperatures led to broken records at many weather stations (see Figure 1, right-hand-side). Unfortunately, this cold event happened a week after an episode of record-breaking high March temperatures also in many places in France and Western Europe (Figure 1, left-hand-side). This sequence led the growing season to start early, with bud

burst occurring in March and the new leaves and flowers left exposed to the deep frost episode that followed.

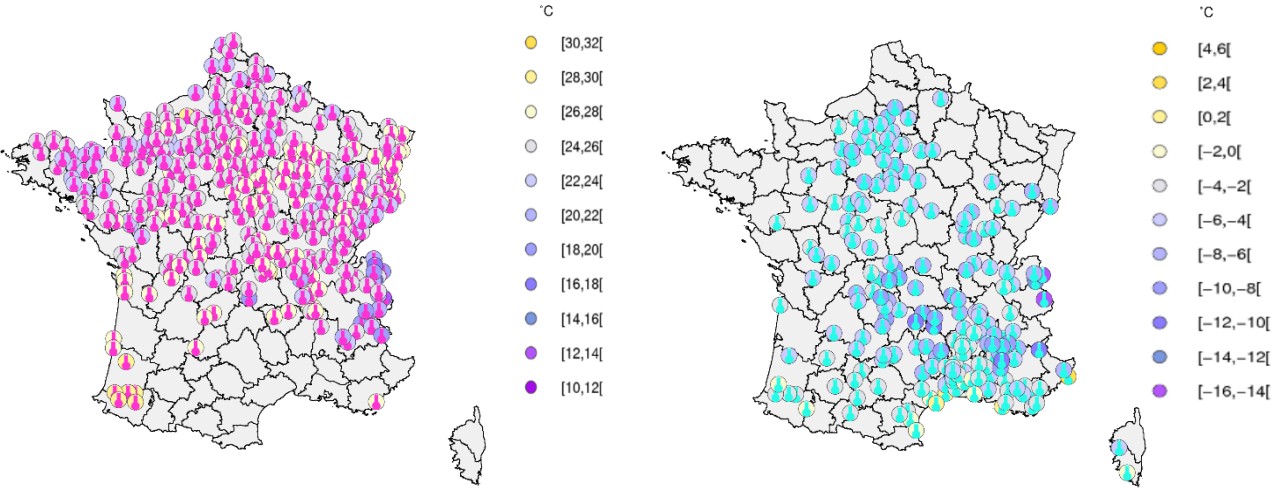

**Figure 1: Stations with March (left) high records broken (pink thermometer) and April (right) low records broken (since at least 20 years) (blue thermometers) in 2021 in France. Symbols are superimposed with the record value of the temperature.**

In 2021, the wine production has been historically low, with 33 bn hectoliters produced, a level that is 25% below the average

production of the previous 5 years, and that is lower than the 2017 production, which was also hit by a late frost (Ministère de l'Agriculture, 2021). Beyond the frost and its consequences, the losses were amplified by a relatively cool and wet summer season allowing Mildew and Botrytis development. In general early varieties in vineyards were affected by frost (for example Sauvignon in Bordeaux). The losses were widespread, but the frost hit the vineyard differently. In hardest hit places such as in Burgundy or Jura, about ⅔ of the production was destroyed. In other places such as in the Beaujolais, later developing species

made the losses less severe. In the Champagne vineyards, and in many places across France the losses ranged from 30% to 50% (Agreste, 2021).





Fruit production was also severely hit for some fruits. Estimates of production losses are of about 50% for pears, cherries, ~25% for peaches and ~20% for apples (with large departures from average depending on the region). Some other productions were also impacted (as sugar beet emergence) but final yields were not finally affected because of favorable production

conditions (Agreste, 2021).

The occurrence of such an event called for investigating the role of climate change. From the weather point of view, the event is rather classical for cold outbreaks, when air masses of polar origin invade Western Europe. The large-scale flow pattern was characterized by a strong high-latitude anticyclone extending from Greenland to the North-Western European coasts, which was found among the 4 most recurrent or stationary North-Atlantic flows (the "Greenland Blocking" pattern of Michelangeli

et al., 1995, and Vautard et al., 1990), inducing a negative value of the North-Atlantic Oscillation (NAO) Index. The combination of polar air advection, cloud-free sky and still long nights led to hours of intense frost. Such dynamical events are not observed to have become more frequent (Screen et al., 2013, Blackport and Screen, 2020) despite the ongoing debate on the role of narrower sea ice extent favoring the occurrence of blocking anticyclones (Barnes and Screen, 2015). The trend in circulation in April is the same as in winter, an intensification of westerly flows that is not related to the weather observed in

2021 (not shown). However, human-induced changes in dynamical conditions, especially leading to cold outbreaks, remain largely uncertain and can be viewed from various indices (Shepherd, 2014), and their understanding would require an in-depth, dedicated analysis.

Here we focus on a statistical analysis of late frosts and their trends from an impact perspective. The exceptional nature of the warm period preceding the 2021 event led to advancing phenology. The advance in the start of the growing season has

increased the number of frost days occurring after the start of the growing season in several places worldwide, including in Europe (Liu et al., 2018). Using several indices for grapevine exposure, it has been found that the date of the last frost day has not regressed as fast as the date of growing season start (Sgubin et al., 2018). However, so far no formal attribution study of a "growing period frost" has been carried out to quantify the role of anthropogenic climate change in these observed trends. This article is devoted to an attribution study of the "growing period frost" event witnessed in April 2021, with a focus on change

in intensities and frequencies of the most extreme cases. It uses several indices characterizing cold temperatures in the growing season, and the well-established attribution methodology described in Philip et al. (2020) and van Oldenborgh et al. (2021).

A rapid attribution analysis was carried out in June 2021 and reported in (Vautard et al., 2021, https://www.worldweatherattribution.org), with several indices developed and analyzed, showing that while spring frosts are generally becoming less severe and frequent, frosts occurring after the growing season start are becoming more intense due to

climate change. Since then, observations were consolidated, more model data has been collected and simulation data processing was homogenized. This article reports the final results, which confirm the conclusions of the preliminary analysis.

We present several definitions of the frost event in section 2, and the corresponding indices chosen. In Section 3, trends in observations are analyzed, and in section 4, results from model ensembles are analyzed. This is followed by a synthesis of results, a discussion and a conclusion in Section 5.



## 2 Event definition and indices used


The cold spell of 6-8 April 2021 hit much of Central and Northern Europe (see Figure 2a). However, we focus here on central/northern France in order to investigate a relatively homogeneous, mostly plain or low-elevation area (see Figure 2b). This area [-1°- 5°E; 46°-49°N] covers most of the grapevine agriculture areas of Champagne, Loire Valley and Burgundy which were identified as specifically vulnerable regions under climate change (Sgubin et al., 2018). The area also covers

regions with high crop and fruit production.

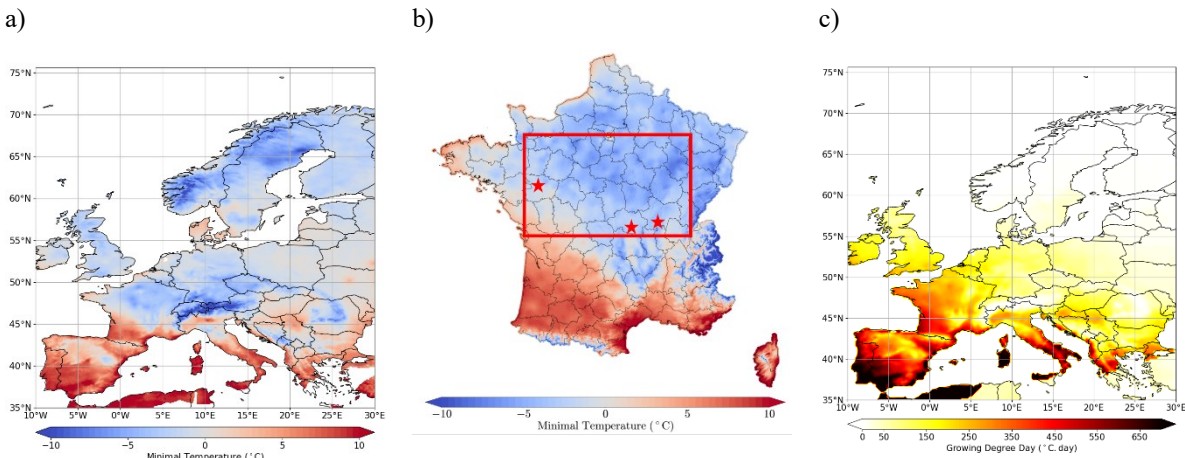

**Figure 2. a) Minimum temperatures on 6 April 2021 in Europe from the E-OBS database (see Section 3); b) focus on France with a higher resolution dataset, using the Anastasia data (Météo-France, Besson et al, 2019). The study area is shown in this panel by the bounded box in red; stars indicate the location of the 3 stations used to assess local trends; c) Spatial distribution of the Growing Degree Day index in Europe on 5 April 2021 as calculated from E-OBS.**

We use several event definitions, accounting for different phenological aspects. Differences in results for these definitions also test the robustness of the attribution. In each case, the "event" is defined as the yearly minimum temperature (TNn) obtained under specific conditions, and then averaged over the area, or taken at specific station locations. A basic reference conditioning is the fixed-season minimum temperature and does not consider phenology: the TNn is calculated over the April-July months (TNnApr-Jul). The second index accounts for phenology. The TNn is calculated conditioned upon the Growing Degree Day

above 5°C (hereafter denoted "GDD") being larger than thresholds characterizing bud burst conditions, which depend on species. In this study, our aim is not to tie thresholds to specific plants' phenology but to provide a general overview for different thresholds. GDD is calculated, at each grid point, with a starting date of the previous winter solstice, in a similar approach used by Garcia de Cortazar-Atauri et al. (2019), assuming that the dormancy break period for grapes is finished in the calculation period. The formula for the GDD at day $t$ during year $y$ is therefore:






$$\text{GDD}(t, y) := \sum_{\tilde{t}=y-1/12/21}^{\tilde{t}=t} \max\left(\text{TM}_{\tilde{t}} - 5, 0\right)$$

(1)

with TM the daily mean temperature and y-1/12/21 is the 21 december of previous year. In 2021, the values of GDD obtained on the day before the frost events in the concerned area vary in the range 150°C.day to 350°C.day, with an average value on 5

April of 259°C.day over the study domain. This value is high for this calendar day (rank=14th since 1921 in the E-OBS extended dataset) but the record value was obtained in 2020, with a mean GDD of 320°C.day. Given the range of values taken in the domain, we considered 3 thresholds for GDD: 250°C.day as a central value, and 150°C.day and 350°C.day as sensitivity experiments. This range of values also helps to capture a range of bud burst values of grapevine cultivars as found in Garcia de Cortazar-Atauri et al. (2009). For each GDD threshold, the yearly minimum TN values (TNn), respectively called hereafter

"TNnGDD250", "TNnGDD150" and "TNnGDD350" for the three GDD thresholds, is calculated over subsequent days and until the end of July at each grid point and then averaged over the domain. Despite the fact that the average characterizes the mean lowest temperature that can occur after crossing the GDD threshold, the average can mix several dates as the GDD threshold crossing and the yearly minimum does not necessarily occur on the same date over the whole domain. In 2021, for instance, the TNnGDD250 was already reached during the 6-8 Apr episode for most of the area, but not in the easternmost

part and in some other parts, because GDD did not exceed 250°C.day during the April frosts.

In order to focus on specific phenological periods when young leaves and flowers are sensitive to frost after bud burst and flowering, we also defined indices over limited ranges of GDD values. The number of possibilities are large, in most cases providing qualitatively similar results. The analysis is reported here only for the range 250-350°C.day. This index is again calculated by grid point before being averaged spatially, or is taken at stations.

Event attribution methods used in this study are well documented in previous studies. The rapid attribution methodology is a classical probabilistic approach, described in Philip et al., 2020 and van Oldenborgh et al., 2021, and has been used in many case studies for heat waves (e.g. Kew et al., 2019, Vautard et al., 2020), extreme precipitation (e.g. Philip et al., 2018), or more complex events such as wildfire weather (van Oldenborgh et al., 2020). It uses a stepwise approach analyzing observations with a Generalized Extreme Value (GEV) with a global warming index as a covariate, then using ensembles of models

validated on the event indices and their extreme value statistics by comparison with observations, and finally using the GEV with the covariate fit to build a statistical model of the data under some assumptions.

In all cases (observations and models), we used data in the 1951-2021 period for the GEV fit. For observations, the covariate is the smoothed observed Global Mean Surface Temperature (GMST), while for models the smoothed Global mean Surface Air Temperature (GSAT) (5-year running average) is used. The only exception is the High Resolution Model Intercomparison

Project (HighResMIP) SST-forced ensemble (see below), for which the observed GMST was used, because of the ensemble forcing.





## 3 Observations and past trends

The observations used here for the attribution are the E-OBS v23e dataset of daily minimum temperatures. In Figure 3 we show the annual time series of the indices, together with trend statistics for the 1951-2020 period. Even though it is displayed

in Figure 3, we did not take into account 2021 in the analysis to avoid selection bias in trend calculation. The Apr-Jul TNn has a slightly upward linear trend of +0.13°C/Decade, which is however not significant at the 90% (two-sided) level because of the large interannual and interdecadal variabilities. By contrast, both TNnGDD250 and TNnGDD250-350 have a significant cooling trend of -0.21 and -0.25°C/Decade respectively. The warming trend in TNnApr-Jul is partly due to larger values since 2000, but these higher values are not reflected in the other indices because GDD also has increased during this period, allowing

lower daily minimum temperatures to be counted earlier in the season. We conclude that, on average, since 1950, extreme yearly minimum temperatures for GDD>250 have cooled by about 1.5-2.0°C. Very low growing-period frosts were also found in 1957 and 1991, with lower values than in 2021.

For different thresholds we also find cooling trends, however with lower significance (Figure 3b). The significance of the signal remains. Interestingly, over the last 50 years (1971-2020) the trends have increased and become more significant (for

instance +0.29°C/Decade, p<0.1 for TNnApr-Jul, and -0.37°C/Decade for TNnGDD250, p<0.1; see also Figure 4).

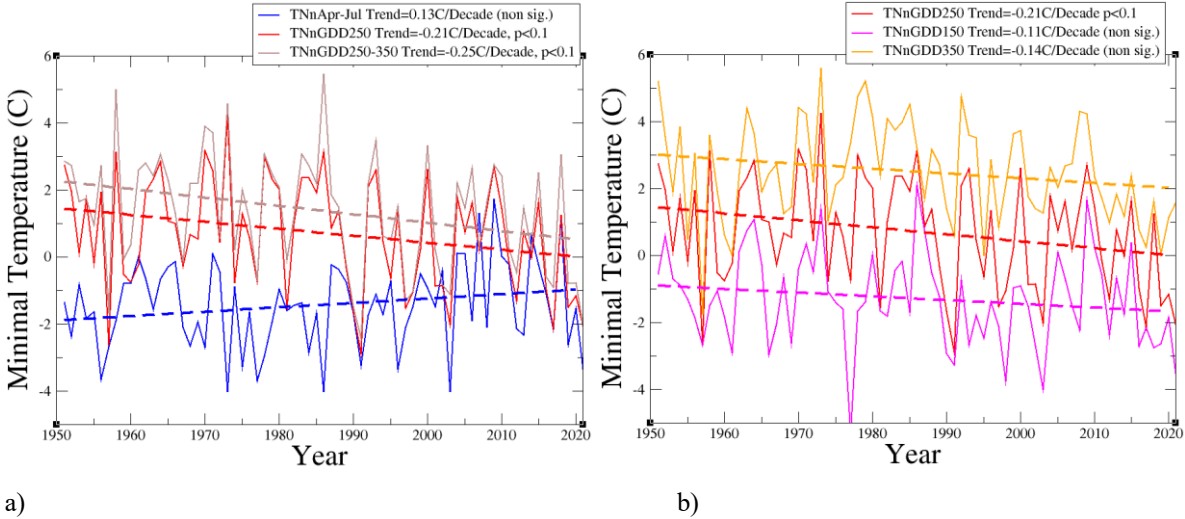

a)                                          b)

**Figure 3: a) Time series of the yearly indices and their respective linear trends calculated over the 1951-2020 period; b) Same as a) but for TNnGDD250, TNnGDD150 and TNnGDD350.**

When considering trends in low extremes of these indices, the results are qualitatively similar but significance is increased when considering GEV fitting using the smoothed observed GMST as covariate instead of assuming a linear trend (see Table 1). We estimate that the event, defined as minimum temperatures over Apr-Jul, has a return period of 78 years [at least 19 years], which means a very rare event in the current climate. However, in a climate corresponding to a global temperature 1.2°C cooler, this would have been about a 1-in-7-year event (best estimate). By contrast, the minimum temperature, taken

over the growing period as characterized by the GDD index, instead of fixed month, has significantly cooled by almost 2°C





with large varying uncertainty ranges and significance depending on the chosen index. The observational analysis is however not sufficient to conclude a role of climate change, which would require models with factual and counterfactual assumptions.

| E-OBS statistics <br> 0 to 1.2° global warming level | TNnApr-Jul | TNnGDD250 | TNnGDD250-350 | TNnGDD150 | TNnGDD350 |
|---|---|---|---|---|---|
| Observed 2021 (°C) | -3.4°C | -2.1°C | -2.0°C | -3.5°C | 1.6°C |
| Return Period 2021 (Yr) | 78 [19;Inf] | 8 [4;25] | 12 [5.0;70] | 9 [4-57] | 2 [1.4-3.2] |
| Return Period -1.2C (Yr) | 7.2 [3.8;19] | 88 [24;inf] | 780 [>53] | 26 [>10] | 9 [3.7;31] |
| Probability Ratio | 0.09 [0;inf] | **11** [2.0;inf] | **63** [>2.3] | 3 [>0.6] | **4.4** [1.3;21] |
| Intensity Change (°C) | **+1.4** [0.2;2.7] | **-2.0** [-3.3;-0.50] | **-2.0** [-3.5,-0.53] | -0.80 [-2.0,0.34] | **-2.0** [-3.4,-0.38] |

**Table 1: Extreme value statistics and observations for the various indices and using the 1951-2020 period and a GEV fit with GMST covariate. Bold font denotes statistical significance.**

To assess the changes at local scale, we also calculated trends for 3 specific stations in the domain (stars in Figure 2). We selected a subset of 3 Météo-France reference stations, which were selected in grapevine regions (Beaucouzé: downstream Loire valley; Charnay-les-Mâcon: Burgundy; Charmeil: Saint-Pourçain grapevine), with several characteristics: for Beaucouzé, light frost and non-exceptional event (-1.3°C) but high GDD (321°C.day on 5 April); for Charnay-les-Mâcon: record frost (-4.4°C, with 266°C.day on 5 April), and for Charmeil: the most severe frost among stations at our disposal (-6.6°C with 244°C.day on 5 April). Detection results are shown in Table 2, for these stations, and for the three main indices: TNnApr-Jul, TNnGDD250 and TNnGDD250-350. In almost all cases, the trends are positive for the fixed season index and negative for the growing season period. However, almost no result is statistically significant. We conclude that at local scale, variability is dominating trend signals (Table 2).


| | Beaucouzé | | Charnay-lès-Mâcon | | Charmeil | |
|---|---|---|---|---|---|---|
| | Value Ret. Per. | PR ΔI | Value Ret. Per. | PR ΔI | Value Ret. Per. | PR ΔI |
| **TNnAprJul** | -1.3°C 11 yr | 0.3 [0.02;1.2] 1.4 [-0.3;3.0] | -4.4°C >100 yr | **0.03** [0;0.9] 1.5 [0.1;2.8] | -6.6°C 85 yr | 0.2 [0.01;7.2] 1.2 [-0.7;3.0] |
| **TNnGDD250** | -1.3°C 5 yr | 1.4 [0.2;9.0] -0.4 [-2.2;1.8] | -4.4°C >50 yr | >1e-4 0.2 [-2;2] | -5.3°C 18 yr | 3.0 [>0.2] -1.0 [-3;1] |
| **TNnGDD250-350** | -1.3°C 7 yr | 1.1 [0.14;7.2] -0.2 [-2.5;2.3] | -4.4°C >90 yr | Infinite 0.3 [-2.0;2.6] | -6.6°C 83 yr | >0.7 -1.5 [-4;1] |

**Table 2: Return periods, probability ratios and changes in intensities obtained from the observations at three stations located as in Figure 2b. Red color indicates a warming change and blue color a cooling change.**

Results here differ from the rapid attribution analysis (Vautard et al., 2021) in the completion and adjustment of the E-OBS

dataset by the producers. This led to slightly different values for the observed indices in 2021. For instance, the estimation of

the TNnGDD250-index based return period was estimated here to 8 years instead of 12 years in the rapid attribution. However

the results are qualitatively similar to those found in the preliminary analysis.

## 4 Models

### 4.1 Model ensembles

For the attribution of the frost event, we use five model ensembles. Each simulation of each ensemble was bias-adjusted using

the CDFt method (Vrac et al., 2016) using the daily minimum and the daily average temperatures from E-OBS over the 1981-

2020 period. Bias correction is an important step here since GDD calculations use a threshold. This method was assessed for

use in climate services in Bartok et al. (2019), and showed good performance. We used statistics of pooled ensembles, using

data until 2021 for the GEV fit of the distributions.

For attribution statistics we used global temperature scaling from the 1951-2020 period and for future trend estimates for a

global warming of 2°C, we used the model GSAT scaling from 2000 to 2050. Change statistics are calculated with respect to

the 2021 year and estimated return period from observation as a reference. As an example, for estimating changes in return

periods (probability ratios) or changes in intensities in model ensembles, for TNnGDD250 the method uses an event with a 8-

year return period in 2021.

The first model ensemble is the Euro-CORDEX (0.11° resolution, EUR-11) multi-model ensemble. It is composed of 75

combinations (as of May 2021) of Global Climate Models (GCMs) and Regional Climate Models (RCMs) for downscaling

(see Vautard et al., 2021 and Coppola et al., 2021 for the description of the ensemble which has increased since these

publications). Each simulation consists of a historical period simulation and a RCP8.5 scenario simulation with fixed aerosol



concentrations. For the attribution of past evolutions historical and scenario are concatenated until 2020. Some simulations
start in 1971, whereas most simulations start from 1951. Given that we need to use data from the previous year for starting
GDD accumulation, all yearly indices are calculated from their second simulation year (i.e. 1972 and 1952 respectively).

The second model ensemble used to study the influence of internal variability was the IPSL-CM6A-LR model (see Boucher
et al., 2020 for a description of the model and Bonnet et al., 2021 for a presentation of the ensemble). It is composed of 32
extended historical simulations, following the CMIP6 protocol (Eyring et al., 2016) over the historical period (1850-2014) and
extended until 2029 using all forcings from the SSP2-4.5 scenario, with the exception of the ozone concentration which has
been kept constant at its 2014 level (as it was not available at the time of performing the extensions).

The third model ensemble is a selection of the CMIP6 historical and SSP3-7.0 simulations. To keep the ensemble balanced we
retained a maximum of three realizations per model. Not all CMIP6 models could be processed at the time of the study. Models
are detailed in the Appendix A, and constitute an ensemble of 45 simulations.

The fourth ensemble used is a set of 10 SST-forced HighResMIP simulations (Haarsma et al. 2016). For the historical time
period (1950-2014), the SST and sea ice forcings used are based on observed dataset, and for the future time period (2015-
2050) the SST and sea ice are derived from CMIP5 RCP8.5 simulations and a scenario as close to RCP8.5 as possible within
CMIP6. The analysis of this ensemble was carried out using the observed GMST as for the observations. The fifth ensemble
is the same set of models run in coupled mode, and the model GSATs were used. Again, more details can be found in the
Appendix A.

The differences with the rapid attribution in models is (i) the homogeneous bias correction, while it was model-dependent in
the rapid attribution, (ii) the addition of the HighResMIP coupled runs, and the change in the CMIP6 selection which was
based on least-biased models instead of bias-corrected models. The present analysis is therefore more consistent across
ensembles.

**4.2 Model evaluation**

We compared the model GEV fit parameters over the overlapping model periods (1951-2020 or 1971-2020) in order to check
the ability of models to simulate such extremes. Such ability was not confirmed for heat waves (eg. Vautard et al., 2020). In
the current case, we found that model ensembles are compatible with the observations accounting for uncertainties (see Table
3) for most indices but not all. Models are said to be compatible with observations when The comparison is made for two
indices for simplicity. For TNnGDD250 the fitted model scale parameter is compatible with the observed one. The shape
parameter is very uncertain in observations, leaving all model fits compatible with them. The same occurs for the TNnApr-
Jul, but in this case all models have an overestimated scale parameter (in terms of amplitude). Only Euro-Cordex and
HighResMIP-SST appear to have a parameter compatible with observations. Given this evaluation, for the final model
"weighted average" (see Philip et al., 2020), only Euro-Cordex and HighResMIP-SST should in principle be considered for
the statistical evaluation of probability ratio and intensity change, while for the TNnGDD250 index, all ensembles can be



considered. However, we have here considered all model ensembles even for the TNnApr-Jul index (see discussion in Section 5).

| Model ensemble / Observation | Index | Scale parameter | Shape parameter |
|---|---|---|---|
| Observation | TNnApr-Jul | 1.18 [0.94;1.38] | -0.24 [-0.42;-0.04] |
| Euro-Cordex | | 1.40 [1.34;1.45] | -0.21 [-0.25;-0.19] |
| IPSL-CM6A-LR | | 1.60 [1.54;1.64] | -0.21 [-0.23;-0.18] |
| CMIP6 | | 1.54 [1.50;1.58] | -0.24 [-0.26;-0.22] |
| HighResMIP-SST | | 1.46 [1.35;1.54] | -0.30 [-0.34;-0.25] |
| HighResMIP | | 1.51 [1.42;1.58] | -0.32 [-0.38;-0.30] |
| Observation | TNnGDD250 | 1.42 [1.12;1.67] | -0.19 [-0.52;+0.07] |
| Euro-Cordex | | 1.52 [1.45;1.56] | -0.24 [-0.26;-0.22] |
| IPSL-CM6A-LR | | 1.54 [1.48;1.59] | -0.26 [-0.29;-0.24] |
| CMIP6 | | 1.61 [1.56;1.65] | -0.21 [-0.24;-0.20] |
| HighResMIP-SST | | 1.55 [1.46;1.64] | -0.23 [-0.29;-0.19] |
| HighResMIP | | 1.50 [1.40;1.58] | -0.24 [-0.31;-0.20] |

**Table 3: Model evaluation, using 2 main indices (TNnApr-Jul and TNnGDD250). Results for TNnGDD250-350 are qualitatively similar to those for TNnGDD250.**


### 4.3 Simulated mean trends

The trends in the two main indices for the 5 model ensembles is analyzed in the form of histograms (Figure 4), in order to examine the variability across ensemble members. There is a large range of minimum temperature trends from April to July, which are almost all positive. The observed trend in the minimum temperature from April to July is close to the median middle

of the distribution for the Euro-Cordex and the CMIP6 ensemble, while it is closer to the lower tail of the distribution of the remaining three ensembles. A large range of possibilities is also found for the trends of the TNnGDD250 index, with a large part of the simulations showing negative and lower trends than those of the minimum temperature from April to May, consistent with the observations. We conclude from these figures that, despite the general trend towards cooling of the growing period frosts, the expected trend, for a given singular member, can also be a warming one, albeit with a smaller chance than

for a cooling one. This large uncertainty also has to be taken into account in any adaptation strategy.


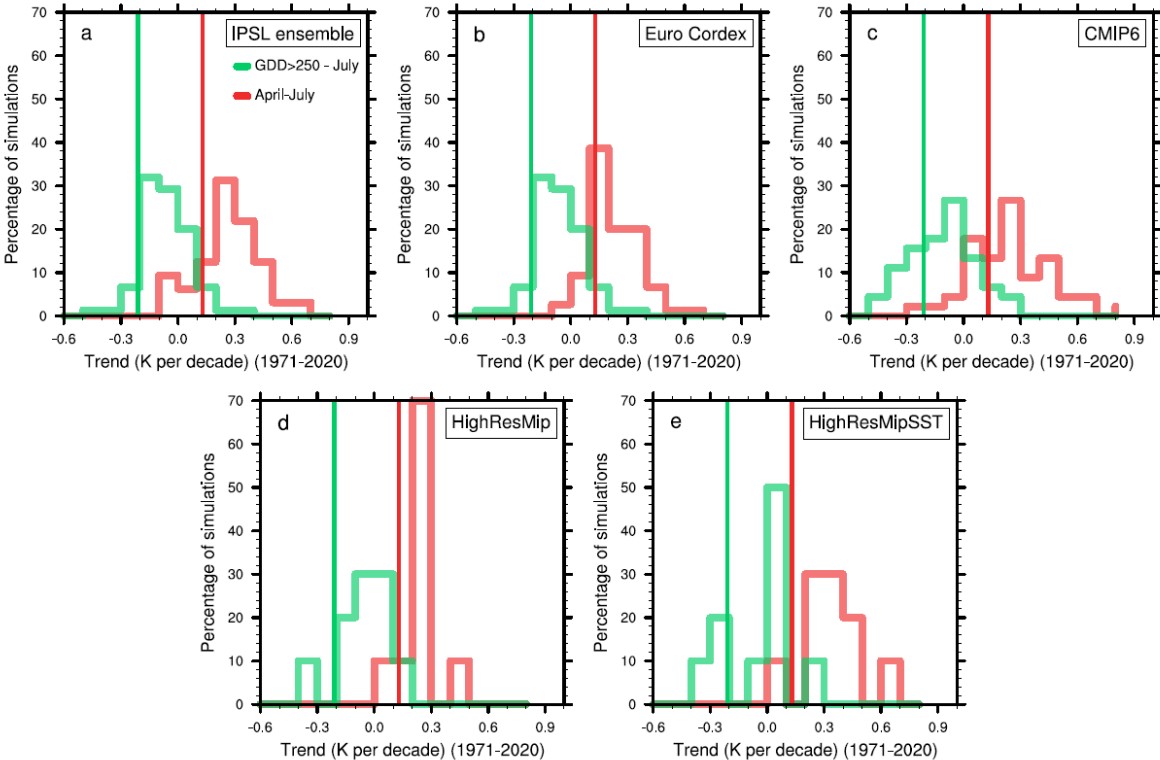

**Figure 4. Histogram of the daily minimum temperature trend calculated from (a) the IPSL ensemble, (b) the Euro-Cordex ensemble, (c) the CMIP6 ensemble, (d) the HighResMip ensemble and (e) the HighResMipSST ensemble (See Section 4.1 and Annex 1 for more details about these ensembles). The observations are represented with the vertical lines. The trends are calculated over the 1971-2020 period for (green) GDD>250 and (red) from April to May.**

## 4.4 Simulated growing period frost extreme trends and attribution

Figure 5 shows, as an example, the change in return values vs. return periods for indices TNnApr-Jul and TNnGDD250 for the Euro-Cordex ensemble, and Table 4 shows the extreme value statistics for all indices for this ensemble as well as other ensembles used. Models show large agreement with observations on changes in return periods and intensities between the preindustrial and current climates for the fixed-calendar TNn index (TNnApr-Jul). The trends in all models seem however underestimated compared to observations for the indices with a GDD conditioning (TNnGDD250).

The behaviour present in all model analysis is illustrated in Figure 5: a clear, significant increase in TNnApr-Jul and an opposite trend sign for the TNnGDD250. Despite being weaker, this increasing trend in low extremes is significant for all ensembles but for HighResMIP-SST (Table 4), with a clear signal of increase in coldest temperatures when considered over the growing period, and with a threshold of 250°C.days. Such a trend is also clear and significant in most ensembles when considering the sensitive range 250<GDD<350 where young leaves and flowers are vulnerable to frost. For the other indices, trends are also significant in most cases but not all.




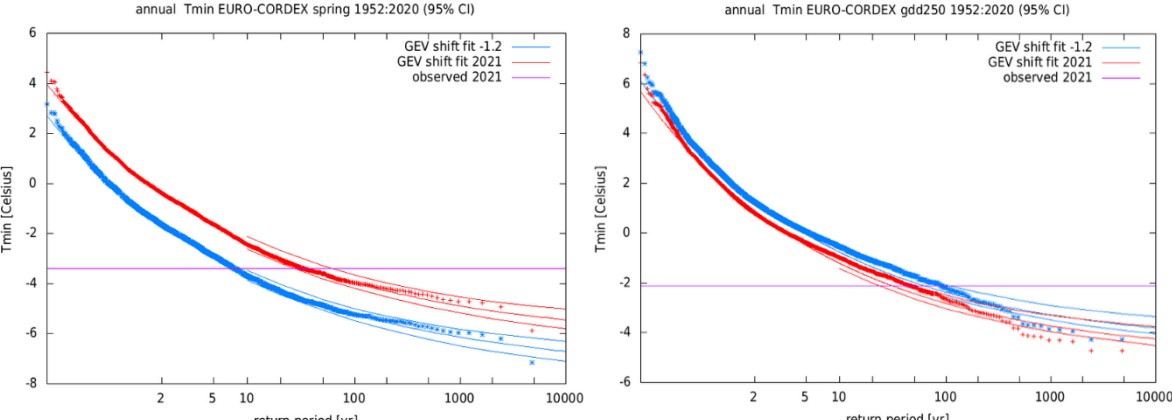

**Figure 5. Return value vs. return period for Euro-CORDEX and the indices TNnApr-Jul (left panel) and TNnGDD250 (right panel). The observed values from E-OBS are marked by the purple line. Note however that the observed values were not the values used to calculate the probability ratio of the event in the Euro-CORDEX ensemble, as the ensemble has a bias toward higher values.**

| Model ensemble / Observation | Index | | Probability Ratio 2021 vs 2021 -1.2°C | Intensity change (°C) 2021 vs 2021 -1.2°C |
|---|---|---|---|---|
| **Observation** | TNnApr-Jul | **RP=78** [19, Inf] | 0.09 [0;inf] | **+1.4** [0.22;2.7] |
| **Euro-Cordex** 2C changes relative to 2021 (+0.8°C) | | 2021 vs p.i. 2C vs 2021 | **0.24** [0.14;0.37] **0.50** [0.29;0.67] | **+1.0** [0.67;1.2] **+0.36** [0.21;0.57] |
| **IPSL-CM6A-LR** | | 2021 vs p.i. | **0.19** [0.13;0.25] | **+1.3** [1.1;1.5] |
| **CMIP6** 2C changes relative to 2021 (+0.8°C) | | 2021 vs p.i. 2C vs 2021 | **0.23** [0.15;0.28] **0.23** [0.12;0.29] | **+1.0** [0.86;1.2] **+0.71** [0.60;0.81] |
| **HighResMip-SST** | | 2021 vs p.i. | **0.07** [0.03;0.16] | **+1.8** [1.2;2.1] |
| **HighResMip** | | 2021 vs p.i. 2C vs 2021 | **0.10** [0.03;0.16] **0.09** [0.;0.17] | **+1.3** [1.0;1.6] **+0.64** [0.54;0.89] |
| **Model average** 2C changes relative to 2021 (+0.8°C) | | 2021 vs p.i. 2C vs 1.2C | **0.18** [0.08;0.37] **0.31** [0.004;2.0] | **+1.2** [0.75;1.7] **+0.58** [0.24;0.92] |
| **Observation** | TNnGDD250 | **RP=8** [4-25] | **11** [2;Inf] | **-2.0** [-3.3, -0.5] |
| **Euro-Cordex** 2C changes relative to 2021 (+0.8°C) | | 2021 vs p.i. 2C vs 1.2C | **1.5** [1.1;1.9] **1.3** [1.1;1.6] | **-0.39** [-0.60;-0.05] **-0.34** [-0.48;-0.07] |
| **IPSL-CM6A-LR** | | 2021 vs p.i. | **1.5** [1.2;2.0] | **-0.36** [-0.61;-0.18] |
| **CMIP6** 2C changes relative to 2021 (+0.8°C) | | 2021 vs p.i. 2C vs 1.2C | **1.4** [1.3;1.7] **1.1** [1.0;1.3] | **-0.39** [-0.54;-0.23] **-0.14** [-0.23;-0.02] |
| **HighResMip-SST** | | 2021 vs p.i. | 1.2 [0.65; 1.8] | -0.21 [-0.58;0.41] |
| **HighResMip** | | 2021 vs p.i. | **2.3** [1.6;3.6] | **-0.73** [-1.1;-0.38] |



| | | 2C vs 1.2C | **1.3** [1.1;1.6] | **-0.21** [-0.41;-0.06] |
|---|---|---|---|---|
| **Model average** | | 2021 vs p.i.<br>2C vs 1.2C | **1.5** [1.1;2.1]<br>**1.2** [1.1;1.4] | **-0.41** [-0.60;-0.22]<br>**-0.20** [-0.30;-0.08] |
| **Observation** | TNnGDD250-350 | **RP=12** [5.0;70] | **63** [2.3;Inf] | **-2.0** [-3.5;-0.57] |
| **Euro-Cordex** | | 2021 vs p.i.<br>2C vs 2021 | **1.7** [1.2;2.7]<br>1.1 [0.98;1.7] | **-0.50** [-0.80;-0.14]<br>-0.14 [-0.47;0.01] |
| **IPSL-CM6A-LR** | | 2021 vs p.i. | **1.9** [1.4;2.8] | **-0.54** [-0.82;-0.29] |
| **CMIP6** | | 2021 vs p.i.<br>2C vs 2021 | **1.5** [1.3;2.0]<br>1.1 [0.99;1.3] | **-0.43** [-0.64;-0.27]<br>-0.09 [-0.23;0.01] |
| **HighResMip-SST** | | 2021 vs p.i. | 1.0 [0.44;1.8] | -0.03 [-0.54;0.69] |
| **HighResMip** | | 2021 vs p.i.<br>2C vs 2021 | **2.8** [2.1;8.9]<br>1.3 [0.93;1.6] | **-0.82** [-1.3;-0.58]<br>-0.21 [-0.36;0.05] |
| **Model average**<br>2C changes relative to 2021 (+0.8°C) | | 2021 vs p.i.<br>2C vs 2021 | 1.7 [0.89;3.2]<br>**1.1** [1.0;1.3] | **-0.50** [-0.94;-0.07]<br>**-0.12** [-0.23;-0.04] |
| **Observation** | TNnGDD150 | **9** [5;82] | 3 [>0.7] | -0.80 [-2.1;0.35] |
| **Euro-Cordex** | | 2021 vs p.i. | **1.3** [1.1;2.0] | **-0.29** [-0.74;-0.12] |
| **IPSL-CM6A-LR** | | 2021 vs p.i. | **1.2** [1.0;1.4] | **-0.28** [-0.46;-0.02] |
| **CMIP6** | | 2021 vs p.i. | **1.4** [1.2;1.6] | **-0.36** [-0.52;-0.19] |
| **HighResMip-SST** | | 2021 vs p.i. | 1.1 [0.61;1.4] | -0.09 [-0.40;0.54] |
| **HighResMip** | | 2021 vs p.i. | **1.7** [1.3;3.9] | **-0.62** [-0.95;-0.21] |
| **Observation** | TNnGDD350 | **2** [1.4;3.2] | **4.4** [1.3;21] | **-2.0** [-3.4;-0.38] |
| **Euro-Cordex** | | 2021 vs p.i. | 1.1 [0.98;1.3] | -0.24 [-0.49;+0.05] |
| **IPSL-CM6A-LR** | | 2021 vs p.i. | **1.2** [1.0;1.3] | **-0.27** [-0.45;-0.07] |
| **CMIP6** | | 2021 vs p.i. | **1.1** [1.0;1.2] | **-0.19** [-0.41;-0.08] |
| **HighResMip-SST** | | 2021 vs p.i. | **1.4** [1.1;1.7] | **-0.54** [-0.94;-0.11] |
| **HighResMip** | | 2021 vs p.i. | **1.4** [1.2;1.8] | **-0.56** [-0.87;-0.21] |

**Table 4: Change in extreme value statistics for all model ensembles and observations, with the GEV model fitted from data over the 1951-2020 period for the past trends estimates, and over the 2000-2050 period for future trends (when estimating the changes for a 2°C warming above pre-industrial levels); We assume here that pre-industrial (p.i.) global warming level is 1.2°C cooler than the 2021 one, and therefore the 2°C warming level is reached when the warming is 0.8°C above the current level. In each row, the values of the probability ratio (the ratio between inverse return periods) are shown as well as the intensity change obtained by using the same return period threshold as in the observations, together with their 95% confidence levels as obtained from a bootstrap estimate using 1000 samples. Numbers in blue indicate a decrease of TN, and in red an increase of TN. The last row indicates changes for a 2°C warming level. Boldface numbers indicate statistical significance against a "no change" assumption.**





Despite a sign agreement between models and observations on trends, models generally simulate much weaker trends for the GDD-conditioned indices than observed, a fact that remains unexplained, just as the underestimation in extreme temperatures in summer heat waves (see eg. Vautard et al., 2020; van Oldenborgh et al., in revision). For TNnGDD250, all ensembles simulate an increase in the frequency of growing-period low extreme temperatures ranging from 10% to 110% with a weighted

best estimate of 50% (see also Section 5). For the other indices the range of factors is rather similar, despite lower values for TNnGDD350. Changes in intensities are also all negative but remain below 1°C.

### 4.5 Future trends

Indices have similar future projected trends as in the past decades in the ensembles and scenarios considered here. Figure 6 shows evolutions of the ensemble-median and 10th and 90th percentiles for Euro-Cordex [RCP8.5] and CMIP6 [SSP3-7.0],

for example, but similar results hold for the other ensembles, which have less members or shorter time coverages. In both cases, the median of April-July minimum temperatures over the region continues to increase with mean values around 2°C while they are below frost level in 2021. By the end of the century, frost such as in 2021 will become a very rare occurrence in April or after in these scenarios. However, frost can still be expected earlier in the year, while at the same time the growing season starts earlier. This can be seen in the development of the TNnGDD250 index throughout the 21st century which shows

a weak decreasing trend. It is noteworthy that in the second half of the century, the 10th percentile often nears or exceeds the 2021 value. More frequent events like the 2021 are therefore expected. By the end of the century for this scenario, we also expect deep frosts in the growing period with intensities which have never been met in 2021 or in earlier years.

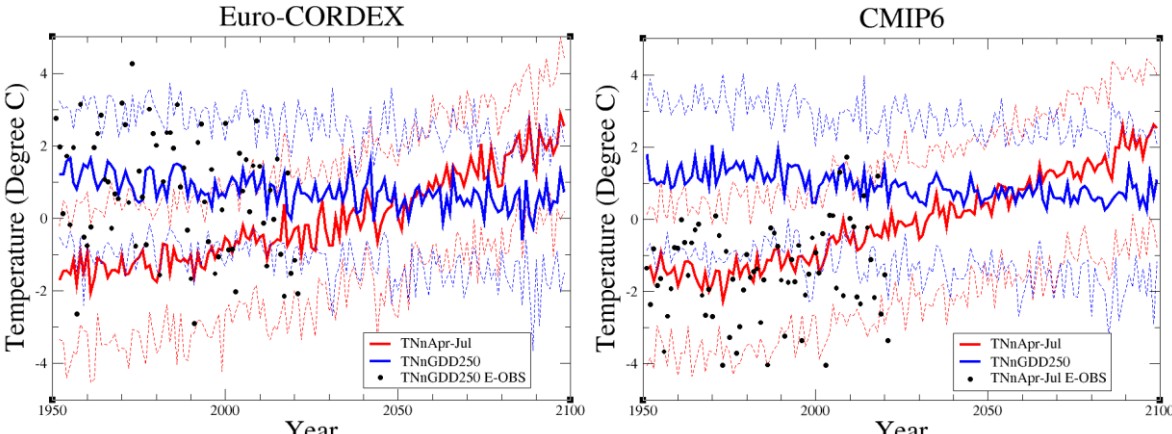

**Figure 6. Time evolution of the median (thick line), and the 10th and 90th percentiles (dashed lines) of the ensembles Euro-Cordex**
**(75 members) and CMIP6 (45 members) for the indices TNnApr-Jul (Red) and TNnGDD250 (blue). Black dots represent the observations from E-OBS, on the left panel for Euro-CORDEX, on the right panel for CMIP6.**

Figure 6 also includes the observed time series for the two indices. For TNnGDD250, even though points generally fit well the 10%-90% model range (expected because the models are bias-corrected), we observe a bias in low extremes with variability



in the observations inducing frequent excursion in temperatures far below the 10% quantile. Such bias is not found in the TNnApr-Jul index.

We restrict the analysis of future trends in extremes to the 2°C warming level above the P.I. conditions, which is assumed to be 0.8°C above current level in 2021. This restriction is made to be on the safe side with potential nonlinearity of response of the extreme indices to global warming while we assume linearity here with the covariate GEV method. In this future case the

GEV fit is carried out over the 2000-2050 period, and probability ratios and intensity changes are given for events with a similar return period as for the 2021 event.

Results are shown along with attribution results in Table 4. Extreme cold temperatures for the April-July period will continue to become less extreme. Euro-Cordex simulations, which are the only ones consistent with observed trends, project that events similar to the 2021 event would become about half as frequent in a 2°C warming climate. The other models predict factors

ranging from between 3 and 10 times less frequent. In contrast, the growing-period extreme frost intensity is increasing, and the 2021 event with a GDD>250 is projected to have an increasing frequency by about 30% [10% - 60%] for a 2°C warmer climate than preindustrial in Euro-Cordex, 10% [0%-30%] for CMIP6 selections and 30% [10% - 60%] for HighResMIP (coupled).

## 5. Synthesis, summary and discussion

The individual assessments described above for probability ratio and intensity changes in the past period are summarized in Figure 7. Given the large differences between models and observations for the growing-period indices TNnGDD250 and TNnGDD250-350, we do not combine the observational and model results to form a single "synthesis" but instead we present the model weighted average for comparison with the observations. In the case of the TNnApr-Jul index, only two ensembles, Euro-Cordex and HighResMip-SST, pass the validation criteria. However, the three additional models (IPSL-CM6A-LR,

CMIP6 and HighResMip) that validate well for TNnGDD250 and TNnGDD250-350, give similar results to the other ones. Incorporating them in the weighted average has no impact on the high significance of the change found, and makes the comparison across indices consistent.

While uncertainties are comparably large for the quantitative assessment of probability ratios there is a significant decrease in the likelihood of cold waves as defined above for TNnApr-Jul. The event that has occurred in 2021, taken as a fixed-season

extreme, has become rare, with a return period of at least 19 years, and with a best estimate of 78 years. The intensity of a cold wave as observed in April is also decreasing, by a well-constrained best estimate of 1.2°C. When considering the lowest temperatures after the growing season start simulated by the GDD thresholds, models and observations quantitatively disagree with respect to probability ratio and intensity, but the qualitative agreement is clear and shows an increase in the likelihood of damaging frost as well as an increase in the intensity across all indices. This is corroborated by the fact that these trends

continue under future warming (see below). This allows a clear qualitative attribution of these trends to anthropogenic climate change with the model results serving as lower bounds.



In Figure 8 we summarize the projected changes in probability and intensity between the present and +2°C climate, showing an unweighted average for the three model ensembles Euro-Cordex, CMIP6 and HighResMIP. We again use all available models for TNn-Apr despite CMIP6 and HighResMIP ensembles not passing the validation over the historical period. We do

so because (i) all models are included for the other two indices and we do not know how well they validate for the future, (ii) no synthesis is formed so the unweighted average shown is only of qualitative use. Probability ratios are less than unity for TNnApr-Jul, indicating that the current trend for decreasing frequency of cold snaps is likely to continue in the future. Projections indicate a decrease by a factor of about 5 in the type of event witnessed in 2021. Likewise, the projections for change in intensity indicate that Apr-Jul cold snaps will continue to warm, by a best-estimated increase of about 0.6°C.

Growing-period minimum temperatures with GDD=250 degree.day continue to decrease with a best estimate of about 0.2°C and an increase in frequency of about 20%.






**Figure 7. Changes between the past and present: summary of observational (blue) and model (red) results for probability ratio (left) and change in intensity [°C] (right) in the three indices TNnApr-Jul (top), TNnGDD250 (middle) and TNnGDD250-350 (bottom). Extent of the bars gives 95% confidence intervals accounting for variability within the data sets and model spread (white) where appropriate, with the black marker indicating the best estimate. A weighted average of model results is shown in bright red. Note that, for the index TNnApr-Jul, only Euro-Cordex passed the validation step but other models are included in the weighted average**
**for reasons described in the text.**





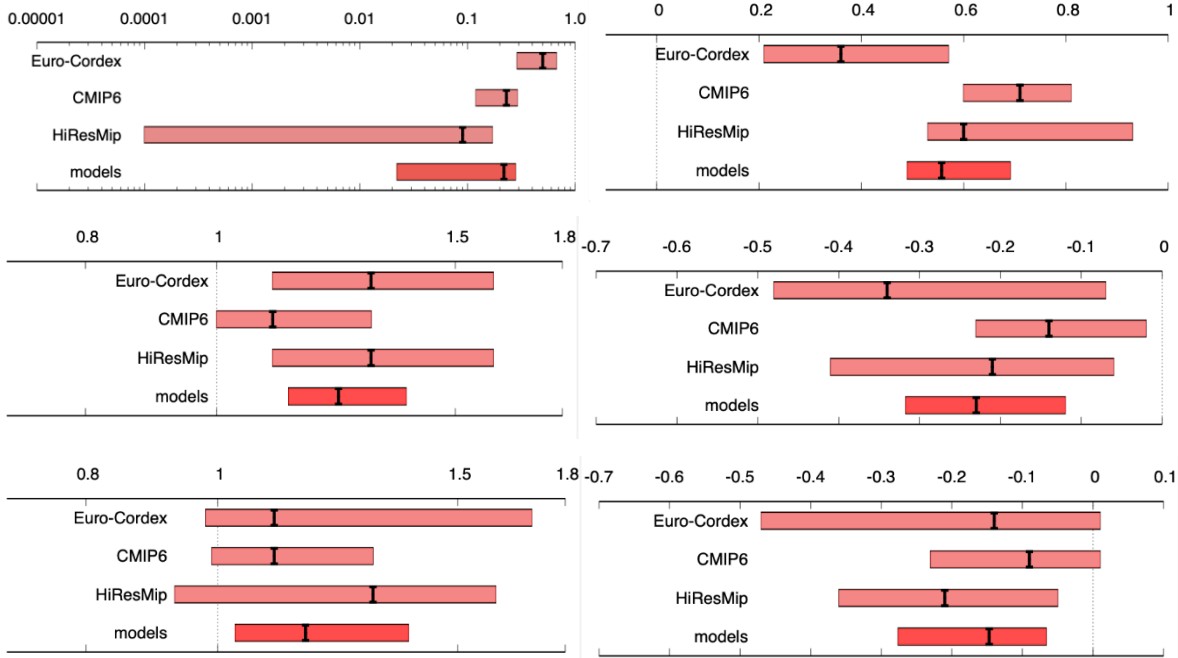

**Figure 8. Projected changes between the present and +2degC climate: summary of results for probability ratio (left) and change in intensity [°C] (right) in the three indices TNnApr-Jul (top), TNnGDD250 (middle) and TNnGDD250-350 (bottom). Extent of the bars gives 95% confidence intervals accounting for variability within the data sets, with the black marker indicating the best estimate. A weighted average of the results is shown in bright red.**

While the growing season is starting earlier, necessary plant dormancy characteristics also change and the lack of chilling winter days may delay the bud burst in many species (Chuine et al., 2016). This effect is not taken into account here and could alter our results concerning changes in bud burst dates. Such dates are also dependent on species. We have tested the dependence on thresholds of a simple GDD index, which provide similar results than the central thresholds discussed in the synthesis. Dormancy effects, as well as other specific plant effects can only be studied through impact models, which was not the goal in this study.

The applicability of our results at local scale is limited in quantitative terms. The local station analysis, and the trends histograms show that given locations are more likely to exhibit cooling of extreme growing-period temperatures than warming, but a warming cannot be excluded at these scales and at present day warming levels.

The discrepancy between trends in models and in observations in the historical periods currently remains unexplained. It shows that either large variability inhibits an accurate estimation of trends of cold extremes or that other factors come into play which may not be well simulated such as trends in radiation or cloudiness as a response to either warming or aerosols. These factors should be investigated in future studies.

Above all, the finding that trends identified up until now continue under future warming indicates that anthropogenic climate change is an important driver of the observed trends and suggests that the models indeed underestimate the effect of change

due to forcing factors and that the discrepancy between observed and simulated trends is not entirely explainable by unmodelled factors other than human-induced climate change.

In conclusion, we identify two key attributable effects, the decrease in likelihood and intensity of minimum temperatures and the increase of likelihood and intensity of minimum temperatures when conditioned on growing degree indices. These findings are consistent across the different lines of evidence pursued despite the quantitative differences. The GDD-indices are however a crude representation of the vulnerability of different species to frost. Thus, our findings highlight that growing season frost

damage is a potentially extremely costly impact of climate change already damaging the agricultural industry but to inform adaptation strategies for specific species impact-based modeling will need to complement our assessment. Other studies, in particular, indicated that impacts may be highly variable across locations and species (Leolini et al., 2018), emphasizing this need.

## Author contribution


RV, GJVO and RB designed the experiments and the related analyses, advised by FO, NV, MR, and IGCA. SL, SK, SP, YR, RB, BD and JMS helped preparing the datasets. All authors contributed to the text.

## Acknowledgements

We are grateful to James Ciarlo (ICTP) who made available in advance to publication on ESGF one of the regional climate

simulations for the Euro-Cordex ensemble. The analysis benefitted from the collection of the Euro-Cordex simulation made within the Copernicus Climate Change Service. The final analysis of this study is partly supported by the European Union's Horizon 2020 research and innovation programme under grant agreement No 101003469 (XAIDA project). It is among the case studies treated in this project. Our colleague co-author Geert Jan van Oldenborgh started the analysis with the author's team but sadly passed away in October 2021 before the article was submitted. The first author dedicates this study to his mother

who passed away on the day before the frost event and who gave him her passion for flowers, nature and weather.

## Annex I. Model ensembles description

This annex provides more details about the model ensembles used in this study.

## 1 EURO-CORDEX

The Euro-Cordex ensemble is made of 75 simulations of 12 Regional climate models downscaling 8 Global Climate Models.

The description of the ensemble is detailed in Vautard et al. (2021) and Coppola et al. (2020), but between the article publication and the start of the study, the ensemble size passed from 55 models to 75 models. The reader is referred to this



publication for a description and an assessment of this ensemble in the historical period. Daily mean and minimum temperatures were corrected at grid point level using the E-OBS observation dataset from 1981 to 2020. Bias correction follows the method described in Vrac et al. (2016) refined in Bartok et a. (2019) and applied on daily data instead of hourly data. The

GCM-RCM ensemble is described in Table A.1 below.

| RCM / GCM | CNRM | EC-EARTH | HadGEM | MPI | NorESM | IPSL | CanESM | MIROC |
|---|---|---|---|---|---|---|---|---|
| CCLM | | | | | | | | |
| HIRHAM | | 3 | | | | | | |
| RACMO | | 3 | | | | | | |
| RCA | | 3 | | 3 | | | | |
| REMO | | | | 3 | | | | |
| WRF361H | | | | | | | | |
| WRF381P | | | | | | | | |
| ALADIN53 | | | | | | | | |
| ALADIN63 | | | | | | | | |
| RegCM | | | | | | | | |
| COSMO-crCLIM | | 3 | | 3 | | | | |
| HadREM | | | | | | | | |

Table A.1: Euro-CORDEX Simulations analyzed in this study. Grey cells indicate a GCM-RCM couple used, and numbers in the cell indicate the number of realizations used (essentially 3 for two of the GCMs).

**2 CMIP6 selected ensemble**

The CMIP6 multi-model ensemble is a set of global climate models, developed by several institutes around the world (Eyring

et al., 2016). Here a subset of CMIP6 models are used, with historical and SSP3-7.0 experiments (Meehl et al. 2014; O'Neill et al. 2014, Vuuren et al. 2014, and O'Neill et al. 2016) together spanning the period between 1850 and 2099 for tas and tasmin variables. The analysis, as for the other ensembles, is however restricted to the years after 1950. Simulations were also bias-corrected but we kept only 3 members maximum per ensemble in order not to overload the results with models having many members. In total, given the available simulations initially, we obtained 45 simulations with models described in Table A.2

below.



| Model | Realization | Model | Realization |
|---|---|---|---|
| ACCESS-CM2 | r1i1p1f1 | INM-CM5-0 | r1i1p1f1 |
| | r2i1p1f1 | | r2i1p1f1 |
| | r3i1p1f1 | | r3i1p1f1 |
| ACCESS-ESM1-5 | r1i1p1f1 | IPSL-CM6A-LR | r1i1p1f1 |
| | r2i1p1f1 | | r2i1p1f1 |
| | r3i1p1f1 | | r3i1p1f1 |
| AWI-CM-1-1-MR | r1i1p1f1 | KACE-1-0-G | r1i1p1f1 |
| | r2i1p1f1 | | r3i1p1f1 |
| | r3i1p1f1 | MIROC6 | r1i1p1f1 |
| BCC-CSM2-MR | r1i1p1f1 | | r2i1p1f1 |
| CanESM5 | r1i1p1f1 | | r3i1p1f1 |
| | r1i1p2f1 | MIROC-ES2L | r1i1p1f2 |
| | r2i1p1f1 | MPI-ESM1-2-LR | r1i1p1f1 |
| CNRM-CM6-1 | r1i1p1f2 | | r2i1p1f1 |
| CNRM-ESM2-1 | r1i1p1f2 | | r3i1p1f1 |
| EC-Earth3-AerChem | r1i1p1f1 | MRI-ESM2-0 | r1i1p1f1 |
| EC-Earth3 | r1i1p1f1 | | r2i1p1f1 |
| | r4i1p1f1 | | r3i1p1f1 |
| EC-Earth3-Veg-LR | r1i1p1f1 | NorESM2-LM | r1i1p1f1 |
| EC-Earth3-Veg | r1i1p1f1 | NorESM2-MM | r1i1p1f1 |
| GFDL-ESM4 | r1i1p1f1 | UKESM1-0-LL | r1i1p1f2 |
| INM-CM4-8 | r1i1p1f1 | | r2i1p1f2 |
| | | | r3i1p1f2 |

**Table A.2: CMIP6 models used in this study, together with the realization when several were available**

**3 IPSL-CM6 single model ensemble**

The IPSL-CM6A-LR model ensemble is a 32-member ensemble of the coupled climate model with the same name. The model is described in Boucher et al. (2020) and the ensemble is presented and evaluated in Bonnet et al., (2021). Simulations start in the pre-industrial period with slightly different initial conditions and are saved in this study for the whole historical period and beyond, until 2029. The ensemble has been used for attribution studies, for instance in the 2019 heatwave attribution described

in Vautard et al. (2020).

**4 HighResMIP SST-forced and coupled ensembles**

We also consider two sets of ensembles from the High Resolution Model Intercomparison Project (HighResMIP, Haarsma et al. 2016), which is a coordinated set of experiments as a part of CMIP6, designed to assess the impact of model horizontal resolution. HighResMIP consists of atmosphere-only (SST-forced) and coupled runs, both spanning 1950-2050. In this study,

we make use of both the SST-forced and coupled ensembles. As briefly described in the main text, in the SST-forced ensemble, for the 'present' time period (1950-2014), the SST and sea ice forcings used are based on the daily, 0.25° x 0.25° Hadley Centre Global Sea Ice and Sea Surface Temperature dataset, with area-weighted regridding used to map this to each model grid; for the 'future' time period (2015-2050), SST/sea-ice data are derived from RCP8.5 (CMIP5) data, and combined with



greenhouse gas forcings from SSP5-8.5 (CMIP6) simulations (interested readers are referred to Section 3.3 of Haarsma et al.
2016 for further details).

Bias correction was performed using the same method as for the other ensembles.

| Model | High | Medium | Low | DOI | Contributed by | Number of simulations used |
|---|---|---|---|---|---|---|
| CNRM-CM6-1-HR | | 720*360 | | https://doi.org/10.22033/ESGF/CMIP6.1387 | CNRM (Centre National de Recherches Meteorologiques), CERFACS (Centre Europeen de Recherche et de Formation Avancee en Calcul Scientifique) (CNRM-CERFACS) | 1 |
| CNRM-CM6-1 | | | 256*128 | https://doi.org/10.22033/ESGF/CMIP6.1375 | CNRM-CERFACS | 1 |
| EC-Earth3P-HR | 1024*512 | | | https://doi.org/10.22033/ESGF/CMIP6.2323 | EC-Earth-Consortium | 3 |
| EC-Earth3P | | 512*256 | | https://doi.org/10.22033/ESGF/CMIP6.2322 | EC-Earth-Consortium | 3 |
| HadGEM3-GC31-HM | 1024*768 | | | https://doi.org/10.22033/ESGF/CMIP6.446 | the Met Office Hadley Centre | 1 |
| HadGEM3-GC31-MM | | 432*324 | | https://doi.org/10.22033/ESGF/CMIP6.190 | the Met Office Hadley Centre | 1 |

**Table A3. Spatial grids of the HighResMIP models in high-, medium,-, and low-resolution groups used in this study, along with**
**relevant references for the simulations, their origins, and the number of simulations used in the analysis**



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
