# Peer review of "Human influence on growing-period frosts like the early April 2021 in Central France"

_Natural Hazards and Earth System Sciences, 2022_

## Author Comment (AC1)

General comments:

This manuscript is devoted to the analysis of an extreme cold spell in early April 2021 that occurred in a vast area of Western-Central Europe, including France. This outstanding event was preceded by an anomalously warm March, both months with record-breaking temperatures of reverse sign in France. The rapid heat accumulation (GDD) until the end of March led to an advancement of phenology (e.g. grapevines and temperate fruit trees), which significantly expose buds/flowers to chilling conditions and late frosts. This two-fold effect (phenology advancement & higher late frost risk) has been reported by several authors, including in studies of viticulture worldwide (a recent revision highlights this issue: doi:10.3390/app10093092). Furthermore, the authors explore, through an attribution analysis using climate model ensembles, the potential relationship between this event and anthropogenic forcing. Opposite trends were identified, either using fixed periods for the minimum temperatures or GDD-based periods, thus highlighting the importance of using heat accumulation as a time frame instead of the conventional DOY. Overall, I found that the results are scientifically sounding but highlight a still very large uncertainty in climate projections, particularly concerning extreme events like late frosts. The data and methods are also adequate for the purposes of the study. The manuscript is generally well written but deserves some improvements as is outlined below. Hence, I recommend its acceptance after some revisions.

*The authors thank the reviewer for the time spent and the thorough review. We will address all points below.*

Specific comments:

1. Fig.1 The symbols are not clear. Please remove the icons within the circles to better render the colour scale.

*We will improve the figure*

2. Ln 68: "The trend..." this sentence is unclear. Please either remove or better explain the opposite relationship between this cold spell and the projected intensification of the westerlies.

*This sentence is removed, as it is not useful in this context.*

3. Fig2c: this panel should be only for France, allowing a better resolution of the target area. Please revise.

*The figure will be improved but we think it is interesting for the readers to see the pattern at a larger scale.*

4. Equation 1: Please edit and improve quality.

*This will be done*

5. Ln 130-141: the attribution method should be more clearly explained for a general reader not familiar with it. Please develop a bit further.

*The text will be improved. Please note that Reviewer #2 asked for restructuring the sections*

6. Ln 155: The reference to Fig. 4 is not appropriate at this stage. It needs some preliminary explanation beforehand.

*This will be done*

7. Tables 1 and 2 are barely explained. Please develop their explanation, as there are several indicators that are not even mentioned in the text.

*This will be done.*

8. Table 2 caption: "Red color indicates a warming change and blue color a cooling change". No colours are shown. Please revise.

*This will be done*

9. Section 4.1: You have used different anthropogenic radiative forcing scenarios in the different model ensembles: SSP2-4.5, RCP8.5, SSP3-7.0, which correspond to very different GHG emissions and concentration pathways. Please explain how these changes may influence your findings. Further, the spatial resolution of the models is not equal. Have you averaged all datasets within the selected domain? I suggest improving and rephrasing this whole section to improve clarity, as several options were taken and they need to be duly justified.

*We use all projections in terms of degree of warming, and arguments showing that this is possible (as in IPCC reports) will be presented.*

*We indeed average all datasets over the selected domain. We will write this more clearly in the event definition section.*

10. Ln 233-236: This paragraph is awkward. You mention that only some model simulations should be considered after the evaluation approach, but you eventually decided to use all of them. Please clarify.

*This is explained in Section 5 but we now clarify also here, the group of sentences is now:*

*"Given this evaluation for this index, for the final model "weighted average" (see Philip et al., 2020), only Euro-Cordex and HighResMIP-SST should in principle be considered for the statistical evaluation of probability ratio and intensity change, while for the TNnGDD250 index, all ensembles can be considered. However, we have here considered all model ensembles even for the TNnApr-Jul index (see discussion in Section 5) for consistency across indices, and because results are qualitatively similar, keeping all models or retaining only the compatible models."*

11. In general, the quality of the figures and tables can be significantly improved. The physical units are not always shown and the resolution is poor, being some of their elements difficult to read. For instance, in Fig.5 caption there is no reference to the geographical area that is being considered. The same applies to other figures. Fig. 7 and 8 are very interesting and informative but difficult to read. I suggest adding labels and an improvement in the captions. Their description in the text can be significantly enhanced to facilitate their interpretation by a larger audience.

*This will be done*

12. English is fine. Only minor spell checking is necessary (e.g., 229 "The").

*This will be corrected*

---

## Author Response (AR1)

**Spring frosts - response to comments in the discussion**

**Reviewer #1**

**General comments:**

This manuscript is devoted to the analysis of an extreme cold spell in early April 2021 that occurred in a vast area of Western-Central Europe, including France. This outstanding event was preceded by an anomalously warm March, both months with record-breaking temperatures of reverse sign in France. The rapid heat accumulation (GDD) until the end of March led to an advancement of phenology (e.g. grapevines and temperate fruit trees), which significantly expose buds/flowers to chilling conditions and late frosts. This two-fold effect (phenology advancement & higher late frost risk) has been reported by several authors, including in studies of viticulture worldwide (a recent revision highlights this issue: doi:10.3390/app10093092). Furthermore, the authors explore, through an attribution analysis using climate model ensembles, the potential relationship between this event and anthropogenic forcing. Opposite trends were identified, either using fixed periods for the minimum temperatures or GDD-based periods, thus highlighting the importance of using heat accumulation as a time frame instead of the conventional DOY. Overall, I found that the results are scientifically sounding but highlight a still very large uncertainty in climate projections, particularly concerning extreme events like late frosts. The data and methods are also adequate for the purposes of the study. The manuscript is generally well written but deserves some improvements as is outlined below. Hence, I recommend its acceptance after some revisions.

**The authors thank the reviewer for the time spent and the thorough review. We address all points below.**

**Specific comments:**

1. Fig.1 The symbols are not clear. Please remove the icons within the circles to better render the colour scale.

**Due to technical reasons, the thermometers could not be removed yet. We work on this issue for the final draft.**

2. Ln 68: "The trend..." this sentence is unclear. Please either remove or better explain the opposite relationship between this cold spell and the projected intensification of the westerlies.

**This sentence is removed, as it is not useful in this context.**

3. Fig2c: this panel should be only for France, allowing a better resolution of the target area. Please revise.

The figure has been rescaled to improve lat/lon aspect ratio. However we think it is interesting for the readers to see the pattern at a larger scale, which exceeds the study domain. Figure 2c also goes with Fig 2a.

4. Equation 1: Please edit and improve quality.

**We have improved the equation**

5. Ln 130-141: the attribution method should be more clearly explained for a general reader not familiar with it. Please develop a bit further.

The text will be improved. Please note that Reviewer #2 asked for restructuring the sections, and the presentation is made here with different sections: we added the description of the new structure at the end of the introduction: "We present several definitions of the frost event in section 2, and the corresponding indices chosen. In Section 3, we present methods, observations and models used, and trends in observations are analyzed, and in section 4, results from observations and model ensembles are analyzed. This is followed by a synthesis of results, a discussion and a conclusion in Section 5". Note also that one table has been changed to a figure. Our response below however refers to the previous verion's tables and figures numbering.

6. Ln 155: The reference to Fig. 4 is not appropriate at this stage. It needs some preliminary explanation beforehand.

**Reference to Figure 4 is removed (it was not necessary)**

7. Tables 1 and 2 are barely explained. Please develop their explanation, as there are several indicators that are not even mentioned in the text.

**The TNnGDD250-350 index is now explained above in the text. The caption of Table 1 now details the content of the table. A few details are also given in the caption of Table 2.**

8. Table 2 caption: "Red color indicates a warming change and blue color a cooling change". No colours are shown. Please revise.

**Colors are now present**

9. Section 4.1: You have used different anthropogenic radiative forcing scenarios in the different model ensembles: SSP2-4.5, RCP8.5, SSP3-7.0, which correspond to very different GHG emissions and concentration pathways. Please explain how these changes may influence your findings. Further, the spatial resolution of the models is not equal. Have

you averaged all datasets within the selected domain? I suggest improving and rephrasing this whole section to improve clarity, as several options were taken and they need to be duly justified.

We use all projections in terms of degree of warming, as in the IPCC report, for instance, which provides the climate change response regardless of the scenario, for short-duration events such as studied here. We added the following in 4.1: "Note that we bring together available simulations which do not follow the same greenhouse gas emission scenarios, which could lead to large difference in climate response for given times. Such would also be the case for individual models' responses. However, this should not be a problem as long as results are compared with fixed degree of warming. Such an approach is also followed by the recent IPCC report where changes in extremes are compared (see IPCC, 2021)."

We indeed average all datasets over the selected domain. We added a sentence: "Indices are calculated exactly as for the observations: model GDD values are calculated at each grid points using Equation (1), and the indices are obtained at grid point level before being averaged over the area of study (rectangle in Figure 2b)."

**Note that Section 4.1 is now partly included in new sections 3.2 and 3.3**

10. Ln 233-236: This paragraph is awkward. You mention that only some model simulations should be considered after the evaluation approach, but you eventually decided to use all of them. Please clarify.

**This is explained in Section 5 but we now clarify also here, the group of sentences is now:**

"Given this evaluation for this index, for the final model "weighted average" (see Philip et al., 2020), only Euro-Cordex and HighResMIP-SST should in principle be considered for the statistical evaluation of probability ratio and intensity change, while for the TNnGDD250 index, all ensembles can be considered. However, we have here considered all model ensembles even for the TNnApr-Jul index (see discussion in Section 5) for consistency across indices, and because results are qualitatively similar, keeping all models or retaining only the compatible models."

11. In general, the quality of the figures and tables can be significantly improved. The physical units are not always shown and the resolution is poor, being some of their elements difficult to read. For instance, in Fig.5 caption there is no reference to the geographical area that is being considered. The same applies to other figures. Fig. 7 and 8 are very interesting and informative but difficult to read. I suggest adding labels and an improvement in the captions. Their description in the text can be significantly enhanced to facilitate their interpretation by a larger audience.

We have improved the figures and captions (Figure 1, Figure 2; captions of Figure 5, 7 and 8 [previous numbering]). We do not see how to add labels to these latter figures.

12. English is fine. Only minor spell checking is necessary (e.g., 229 "The").

We have corrected the truncated sentence.

**Reviewer #2**

The authors analyze the human influence on the harsh frost weather in early April 2021 after a unusually warm March over central France, which pose great damages on the grapevine and fruit trees. The results show that human-induced climate change has significant impacts on such extreme events. The topic is interesting and the workload is also relatively heavy. However, I think there is a framing issue and some methodological issues, which may pose a question about the coherency and credibility of the study.

**We thank the reviewer for the effort and constructive comments. We address all comments below.**

At first sight, I think it is a attribution study according to the title. But when I read the whole paper, I feel like it is a evaluation and projection study. Thus, I think the authors should first clarify the main purpose and storyline. In terms of the framing, it is usual to first describe data and methods used in one paper, and then elaborate the results. The results can be organized as observation, model evaluation and attribution. In this paper, the data and methods are combined with the results, which add difficulties to reading and understanding. This study used five model ensembles with different resolutions, and the future scenarios include RCP8.5 and SSP2-4.5. But what is the purpose to use all of them ?

It is true that when carrying out an attribution study, it is necessary to evaluate the models, especially regarding the specific aspects of the extreme event in question. This is why such model evaluation is performed each time. And in addition to the attribution itself, we often add future projections of probability change for the models, which brings important information for future adaptation. To account for this remark we will add a paragraph in the introduction explaining this.

Regarding the different model ensembles and scenarios, we can not say one model or scenario is the best, they are all possible. Taking different scenarios is necessary to explore the possible evolution of climate related to mitigation options, whereas taking several models is necessary to consider the uncertainty in our knowledge on the climate system. However, here we use different scenarios together in a way that results are comparable across them, by using global warming levels as references for changes, in a similar approach as the IPCC report has done in 2021.

Single models usually do not give a reliable description of the probability distribution of trends. To span the range of possibilities and to get an indication of the model uncertainties we use as many models that pass the validation tests and as many scenarios as possible.

Regarding the article structure itself, we follow the reviewer's remark: we clarified the goals and restructured it by presenting first the definition of the event, then presenting all data, methods, observations and models, and then describing the results. The paper involves too many unclear and inaccurate descriptions as well as the inappropriate choices of the methods. The following is a list of some specific comments:

(1) In terms of model evaluation, the Kolmogorov–Smirnov nonparametric test is often applied to determine whether two probability distributions are well-distinguished. In addition, the observed and simulated time series can also illustrate whether the models have the ability in reproducing the observation.

Indeed, a KS-test would probably be interesting for checking the agreement between a model and observations. This would require a comparison with the method of Philip et al, 2020 which only compares inferred coefficients in order to validate a model. Since this paper is about the attribution of a particular extreme event, and not the methodology used, we prefer to use the already validated method of Philip et al 2020, and test this approach in another research paper.

Philip, S., Kew, S., van Oldenborgh, G. J., Otto, F., Vautard, R., van der Wiel, K., King, A., Lott, F., Arrighi, J., Singh, R., and van Aalst, M.: A protocol for probabilistic extreme event attribution analyses, Adv. Stat. Clim. Meteorol. Oceanogr., 6, 177–203, https://doi.org/10.5194/ascmo-6-177-2020, 2020.

(2) There are too many data tables. It is more visual to draw figures like boxplot or bars.

There are currently 4 data tables. We have transformed one of the tables into boxplots (the third for evaluation).

Note: in the initial response we thought we would replace 2 of the tables, but ex Table 2 is actually hard to transform into a figure due to multiple parameters to show.

(3) Section 4.5 is not necessary in this paper, and the results derived from two ensembles with different scenarios are not comparable. The author can do more literature research and write another projection paper.

We emphasize the importance of providing future assessments as an extension of the attribution, as this uses the same data and methods and provides a quick overview of how probabilities will change in the future, offering important insights for future adaptation, hence we prefer to keep it here, even though it is a small section.

We don't know which scenario will occur. But as we use the different scenarios and analyse results for the same warming level of climate change rather than for a given year we can compare these results. This is explained in an added sentence: "Note that we bring together available simulations which do not follow the same greenhouse gas emission scenarios, which could lead to large difference in climate response for given times. Such would also be the case for individual models' responses. However, this should not be a problem as long as results are compared with fixed degree of warming. Such an approach is also followed by the recent IPCC report where changes in extremes are compared (see IPCC, 2021)."

(4) The paper only focus on the trend of regional mean, and it is insufficient to know the pattern distributions.

The paper focuses on regional mean but also provides some results for individual stations. For stations, results for the GDD indices bear large uncertainties as can be seen from Table 2. We think patterns within the domain will not be significant and interpretable due to uncertainties. Considering patterns at a larger scale, for areas with different climatologies, would drive the analysis too far from the event itself, its impacts, which could be done in another paper but is beyond the scope of this current study.

---

## Author Response (AR2)

**Human influence on growing-period frosts like the early April 2021 in Central France**

Finale modifications in production files

5
1. Quality of Figure 1 has been improved.
2. A section Data / Code availability is now added